# Who Is the Most Vulnerable to Anxiety at the Beginning of the COVID-19 Outbreak in China? A Cross-Sectional Nationwide Survey

**DOI:** 10.3390/healthcare9080970

**Published:** 2021-07-30

**Authors:** Bei Liu, Bingfeng Han, Hui Zheng, Hanyu Liu, Tianshuo Zhao, Yongmei Wan, Fuqiang Cui

**Affiliations:** Department of Laboratorial Science and Technology & Vaccine Research Center, School of Public Health, Peking University, Beijing 100191, China; 18827380717@163.com (B.L.); hanbingfeng@pku.edu.cn (B.H.); zhenghui@chinacdc.cn (H.Z.); liuhanyuu@bjmu.edu.cn (H.L.); ztshuoshuo@163.com (T.Z.); wanyongmei76@163.com (Y.W.)

**Keywords:** COVID-19, anxiety scores, population, psychosocial, China

## Abstract

(1) Background: The COVID-19 pandemic has not only changed people’s health behavior, but also induced a psychological reaction among the public. Research data is needed to develop scientific evidence-driven strategies to reduce adverse mental health effects. The aims of this study are to evaluate the anxiety reaction of Chinese people and the related determinants during the earliest phase of the COVID-19 outbreak in China. Evidence from this survey will contribute to a targeted reference on how to deliver psychological counseling service in the face of outbreaks. (2) Methods: A cross-sectional, population-based online survey was conducted from 28 January to 5 February 2020 using an open online questionnaire for people aged 18 years or above, residing in China and abroad. The socio-demographic information of the respondents was collected, and anxiety scores were calculated. A direct standardization method was used to standardize anxiety scores and a general linear model was used to identify associations between some factors (e.g., sex, age, education, etc.) and anxiety scores. (3) Results: A total of 10,946 eligible participants were recruited in this study, with a completion rate of 98.16% (10,946/11,151). The average anxiety score was 6.46 ± 4.12 (total score = 15); women (6.86 ± 4.11) scored higher than men (5.67 ± 4.04). The age variable was inversely and significantly associated with the anxiety score (β = −2.12, 95% CI: −2.47–−1.78). People possessing higher education (β = 1.15, 95% CI: 0.88–1.41) or a higher awareness of cognitive risk (β = 4.89, 95% CI: 4.33–5.46) reported higher levels of anxiety. There was a close association between poor subjective health and anxiety status (β = 2.83, 95% CI: 2.58–3.09). With the increase of confidence, the anxiety of the population exhibited a gradual decline (β = −2.45, 95% CI: −2.77–−2.13). (4) Conclusion: Most people were vulnerable to anxiety during the earliest phase of the COVID-19 outbreak in China. Younger women, individuals with high education, people with high cognitive risk and subjective poor health were vulnerable to anxiety during the epidemic. In addition, increasing confidence in resisting this pandemic is a protective determinant for individuals to develop anxiety. The findings suggest that policymakers adopt psychosocial interventions to reduce anxiety during the pandemic.

## 1. Introduction

COVID-19 has spread worldwide and created an unprecedented crisis [1]. The World Health Organization (WHO) declared the pandemic as a Public Health Emergency of International Concern on 30 January 2020 [2]. As of 28 January 2020, when this study was conducted, the COVID-19 virus had infected 5974 people in Mainland China; as of 1 February 2020, there were 14,380 cases of infection that had been reported [3]. As the novel coronavirus was highly contagious in nature, it created global fear. By the end of June 2020, more than ten million confirmed COVID-19 cases had been detected in 216 countries, territories, and areas, and more than 500,000 deaths had been reported [4]. The large number of patient deaths and stressful situations caused by the pandemic led to warnings being issued to society. 

Anxiety in the general population is currently a major public health concern [5]. Widespread anxiety and distress can cause serious social and economic disruption during pandemics [6]. Negative emotions can cause behavioral changes such as being afraid to leave the house and continuously disinfecting the environment [7]. The lessons learned from experiences with the severe acute respiratory syndrome (SARS) [8,9,10], pandemic influenza A (H1N1) [11,12,13], and influenza A (H7N9) [14,15] in various culture settings demonstrate that attention must also be paid to mental health as part of the COVID-19 epidemic prevention and control. Cultural differences, disease perceptions, government involvement, and the stage of the outbreak are associated with public response, and these factors vary by disease and settings [16,17,18]. In the early phase of the pandemic, the lack of knowledge on the virus and the absence of valid information easily caused universal anxiety and panic among the general population [19]. Examining the general population’s psychological response during the initial phase of an emerging epidemic is useful for keeping both policy makers and the public informed about the state of preparedness. With COVID-19 contributing to increasingly difficult circumstances and amplified grief reactions, many individuals are prone to experiencing more mental health and psycho-social problems [20]. An indefinite period coupled with stressful situations lead to anxiety in people, which is exacerbated by internet rumors and misinformation about the pandemic. In fact, the onslaught of this “infodemic” has led to greater fear and worry among the population [21]. Thus, pessimism towards any kind of information can have negative psychological effects, easily leading to public cognitive risk, which is an inherent feature in all human cognitive activity, as well as an index that can measure the psychological panic of the public. Previous studies have suggested that people with higher risk perceptions were more likely to take comprehensive precautionary measures against infection [21,22]. At the same time, risk cognition also affects public psychology states. Excessive risk cognition increases the likelihood of an array of negative emotions, including anxiety and panic [23]. 

People worldwide may be particularly vulnerable to the adverse mental health effects caused by a lockdown, shielding, self-isolation, and physical distancing measures due to the COVID-19 pandemic. Anxiety, worry, and panic increased and became widespread during the epidemic and remained high in the post-outbreak period. Previous studies on SARS and H1N1 in different countries suggested that widespread anxiety and distress occurred in both the affected areas and the overall population [6]. Recent studies have reported that symptoms of anxiety and depression (16–28%) and self-reported stress (8%) are common mental issues during the COVID-19 pandemic [24]. Other research has examined different fields of mental health such as population anxiety, the psychological impact of quarantine, anxiety in medical workers fighting COVID-19, and anxiety caused by countrywide quarantine [25,26]. Subsyndromal mental health problems are a common repercussion during the COVID-19 pandemic [27]. There is accumulating evidence suggesting that investigating the level of anxiety in individuals and identifying the factors of anxiety can help scholars and practitioners clearly comprehend the severity of the pandemic’s effect and improve the effectiveness of health risk communications [28,29]. The aim of this study is to evaluate the anxiety reaction of Chinese people and the related determinants during the earliest phase of COVID-19 outbreak in China. Targeted interventions and psychological consultation services can thus be provided based on the scientific evidence for the target population. 

## 2. Materials and Methods

### 2.1. Study Population

A cross-sectional, population-based online survey was conducted from 28 January to 5 February 2020. It was an open online questionnaire for people aged 18 years and above, residing in China and abroad. Everyone who saw it and was willing to respond could complete the questionnaire using mobile phones or computers.

### 2.2. Measures

We designed a structured Chinese questionnaire and collected data on Wenjuanxing, an online platform providing functions equivalent to Amazon Mechanical Turk. Through the questionnaire, we collected the following information: (1) the socio-demographic information of the respondents; (2) anxiety reaction towards COVID-19; (3) subjective health; (4) awareness of cognitive risk; and (5) confidence in combatting the COVID-19 pandemic. 

#### 2.2.1. Socio-Demographic Variables 

The demographic information collected included age, sex, marriage, education, occupation, area/province, family members, residence. Contact history variables included close contact with an individual with confirmed COVID-19, indirect contact with an individual with confirmed COVID-19, and contact with an individual with suspected COVID-19 or infected materials. 

#### 2.2.2. Anxiety Reaction towards COVID-19 

Participants’ anxiety reaction was measured via 5-item short forms of the State scale of the Spielberger State–Trait Anxiety Inventory (STAIS-5) and adjusted to adapt to the Chinese context [30]. Participants answered each item on a 4-point scale (from 0 to 3 points). The total anxiety score was divided into normal (0–6), mild anxiety (7–9), moderate anxiety (10–13), and severe anxiety (14–15). Someone scoring ≥10 on the STAIS-5 should be considered potentially clinically anxious [30]. The internal reliability (α) was 0.877. 

#### 2.2.3. Subjective Health Status 

Subjective health status was measured via one item: “How would you define your health status?” Health status was divided into four categories, ranging from 1 to 4, with 1 = unhealthy, 2 = ordinary, 3 = good health, and 4 = very healthy.

#### 2.2.4. Cognitive Risk 

Cognitive risk was assessed based on previous studies conducted among the general public [31], with one item examining how likely participants thought it was that they would contract the virus: “How likely do you think it is that you will get COVID-19?” Risks were divided into five categories ranging from 1 to 5, with 1 = no risk, 2 = low risk, 3 = medium risk, 4 = high risk, and 5 = extremely high risk. 

#### 2.2.5. Confidence 

Participants’ confidence was measured via one item: “How confident are you about combatting the COVID-19 pandemic?” Confidence was divided into five categories, ranging from 1 to 5, with 1 = very unconfident, 2 = unconfident, 3 = somewhat confident, 4 = confident, 5 = very confident. 

### 2.3. Data Management and Statistical Analysis

We used SPSS (version 20.0, IBM, New York, NY, USA) and STATA (version 15.1, Stata Corp LLC, College Station, Texas, TX, USA) for data cleaning and statistical analysis. Categorical variables were expressed as absolute and relative frequencies in different groups. We standardized anxiety scores to improve comparability among provinces by adjusting for age and education. Tests comparing demographic variables among anxiety score categories were performed using a two-tailed *t*-test for continuous variables and the Pearson c^2^ test for categorical variables. The general linear model (GLM) was used to analyze associations between socio-demographic factors, cognitive risk, confidence, and anxiety scores, adjusted respectively. The β values and their 95% confidence intervals (CI) were calculated as estimates of the correlations. All *p* values were 2-sided and *p* < 0.05 was considered statistically significant. A dose-response analysis with curve fitting was conducted using Empower (R) (Empowerstats.X&Y solutions Inc., Boston, MA, USA). The spatial data analyses were conducted using ArcGIS (version 10.2, ESRI Corp, Redlands, CA, USA). 

### 2.4. Quality Control

We monitored the progress of the survey daily. After the collection, we checked the accuracy of data, and excluded the questionnaire if (1) the age range was below 18; (2) the answering time was less than 150 s; and (3) there was logical contradiction between the answers to the questionnaire. All data were checked for consistency by two members.

### 2.5. Ethical Approval

This study was approved as an ethical exemption by the Peking University Health Science Center Ethics Committee (IRB00001052). All subjects participated in the survey voluntarily, and the information in the database was completely de-identified.

## 3. Results 

### 3.1. Participants and Characteristics

A total of 11,151 individuals participated in this online survey. Among these, 205 were excluded due to the fact that they were out of the age range or provided incomplete questionnaires, and the rate of completeness was 98.16%. Among the 10,946 eligible participants, 176 (1.61%) were from Hubei province, 10,552 (96.40%) were from other provinces in China (mainly from Beijing, Shandong, Sichuan, Hainan, Guangxi, etc.), and 218 (1.99%) were from abroad (Figure 1). Table 1 shows the socio-demographic characteristics of the participants.

We found that nearly half of the participants had varying levels of anxiety; 4.17% (457) had severe anxiety, 23.82% (2608) had moderate anxiety, and 21.84% (2391) had mild anxiety. Tests of the group differences through multivariate analyses of variance revealed the following: participants with severe anxiety were more likely to be younger female, unmarried, have a bachelor’s degree or higher, and have a less healthy condition as compared to those with mild or moderate anxiety (*p* < 0.001) (Table 1). Based on the standardized anxiety scores, we found that people in Hubei province, which was the epicenter of this disease in China, were the most anxious, followed by those living in Shanghai, Beijing, and Zhejiang (Figure 2). 

### 3.2. The Association between Age and Anxiety Score

Table 2 shows the association between age and anxiety score from the GLM analysis. As age increased, the anxiety of the population gradually decreased in the sex/marriage-adjusted model (β = −2.15, 95% CI: −2.46–−1.84). Anxiety was highest in those younger than 30 years old. In addition, there was a close relation of age to anxiety scores after adjusting for other factors affecting anxiety in a multivariate model (β = −2.12, 95% CI: −2.47–−1.78). Overall, the age variable was inversely and significantly associated with the anxiety score. The *p* value for trend was < 0.001.

In the analyses stratified by hierarchy (Appendix A
Table A1), the results show that the anxiety score of females was much higher than that of males. An inverse association of age with the anxiety score was consistently present in the sex hierarchies.

### 3.3. The Association between Education and Anxiety Score

The association between the educational qualification of the participants and the anxiety score from the GLM analysis is demonstrated in Table 2. Compared to the senior high school and below group (17.64%), the bachelor’s degree (57.11%) and master’s degree or above groups (25.25%) showed a closer connection with the anxiety score (*p* < 0.001). After adjustment for age, sex, and marriage, it was noticed that people with higher education were more anxious about the outbreak (β = 1.19, 95% CI: 0.92–1.46). In addition, there was a close link between education and the anxiety score after adjusting for other factors affecting anxiety in a multivariate model (β = 1.15, 95% CI: 0.88–1.41). Overall, the education variable was positively and significantly associated with the anxiety score. The *p* value for trend was < 0.001.

### 3.4. The Association between Health and Anxiety Score

The association between health and anxiety scores from the GLM analysis is shown in Table 2. Compared to those in a very healthy condition (57.85%), people who were ordinary or unhealthy (10.20%) were more anxious about the outbreak in the model adjusted for age, sex, and marriage (β = 2.78, 95% CI: 2.52–3.03). Furthermore, health condition was associated to the anxiety score after adjusting for other factors affecting anxiety in a multivariate model (β = 2.83, 95% CI: 2.58–3.09). Overall, it could be said that the health variable was significantly associated with the anxiety score. The *p* value for trend was < 0.001.

### 3.5. The Association between Cognitive Risk and Anxiety Score

The association between cognitive risk and the anxiety score from GLM analysis is reflected in Table 3. Compared to the no cognitive risk group (16.01%), the high cognitive risk (5.40%) and extremely high cognitive risk groups (1.62%) showed a close connection with the anxiety score (*p* < 0.001). After adjustment for age, sex, and marriage, people with higher cognitive risk were more anxious about the outbreak (β = 5.15, 95% CI: 4.57–5.73). Additionally, extremely high cognitive risk was connected with anxiety score after adjusting for other factors affecting anxiety in a multivariate model (β = 4.89, 95% CI: 4.33–5.46). Overall, the cognitive variable was positively and significantly associated with the anxiety score. The *p* value for trend was < 0.001.

### 3.6. The Association between Confidence and Anxiety Score

The association between confidence and the anxiety score from GLM analysis is demonstrated in Table 3. With the increase in confidence, the anxiety of the population gradually decreased in the model adjusted for age, sex, and marriage (β = −2.92, 95% CI: −3.26–−2.58). Additionally, it was noticed that confidence was connected with anxiety scores after adjusting for other factors affecting anxiety in a multivariate model (β = −2.45, 95% CI: −2.77–−2.13). Overall, the confidence variable was inversely and significantly associated with the anxiety score. The *p* value for trend was < 0.001.

### 3.7. The Dose–Response Relationship of Age, Education, and Anxiety Score

Figure 3 shows that the relationship between age and the anxiety score is nonlinear. The risk of anxiety decreased with increasing age, as shown by the estimated dose–response curve. Appendix A
Figure A1 shows that the relationship between education and the anxiety score is nonlinear. The risk of anxiety increased with higher education, as shown by the estimated dose–response curve.

## 4. Discussion

Our web-based study indicated that nearly half of the participants experienced varying degrees of anxiety, with 4.17% (457) experiencing severe anxiety, 23.82% (2608) experiencing moderate anxiety, and 21.84% (2391) experiencing mild anxiety. The proportion of anxiety reported in this study was higher than that of Iran [32], and similar with that of India [33]. Another survey supported the same viewpoint—that people’s psychological responses to COVID-19 were dramatic during the outbreak in China [34]. Our study demonstrated that nearly 7.02% (772) of the participants had high risk perception, and nearly 88.72% (9711) of them were confident that the government could control the outbreak. People of Hubei province in China, the epicenter of this disease, were the most anxious, followed by those living in Shanghai, Beijing, and Zhejiang, which are some of the most economically prosperous areas in China. These prosperous areas had the most interaction with Hubei, and in turn received a large number of infected travelers returning from Wuhan during special holidays such as the Chinese New Year. This was also the reason why the most high-risk areas during the outbreak were no longer Guangzhou, Shanxi, Hebei, Tianjin, and Jilin, although residents of those provinces were the most seriously affected during the SARS outbreak. In the early stage of the epidemic, travel data identified cities and regions susceptible to potential future outbreaks. People living in cities with large population mobility were found to be more prone to anxiety, restlessness, depression, and fear [35].

Our results showed age to have a significant inverse association with the anxiety score. With increasing age, the anxiety score of the population gradually decreased in the multivariate model (β = −2.12, 95% CI: −2.47–−1.78). The results are consistent with another survey [27]. A higher anxiety reaction could be triggered in younger populations. The COVID-19 pandemic created a huge challenge for the elderly [36]. However, younger people who had easier access to plentiful channels of information [27] had an increased anxiety reaction in the face of the emerging epidemic [21]. In sex hierarchies, our results showed that the anxiety level in females was much higher than that of males. Young females were the most vulnerable to anxiety, which is consistent with results reported during the SARS outbreak [37] and the H1N1 pandemic [13]. Similar findings in the Iranian general population were reported during the COVID-19 pandemic [32]. In recent years, with the world moving faster and increasingly becoming more competitive, young females have been experiencing increasing pressure from work, education, and life. In addition, females are much more perceptive and vulnerable to their surroundings [38]. One previous study showed that females were more likely to express negative emotional responses and exhibit avoidance behaviors in response to the avian flu [39]. Thus, it can be deduced that during an epidemic, vulnerable young females should be provided with more psychological support or counseling service to maintain their mental health and well-being [27].

The education variable was positively and significantly associated with anxiety scores in the multivariate model (β = 1.15, 95% CI: 0.88–1.41). Our findings demonstrated that anxiety scores were higher in people with higher levels of education, which reflected similar findings in other studies [40,41]. In most cases, those with higher education, who were more likely to possess prior knowledge and experience regarding the risks of an outbreak of infectious diseases, had experienced the SARS epidemic, especially in China. Additionally, individuals with higher educational degrees may be more capable of recognizing the risks of COVID-19 and also have a higher risk perception [42], leading to extended psychological anxiety during the initial stages of the outbreak [43,44]. In fact, at that stage, the occurrence of human-to-human transmission was rampant. This pandemic has caused 10 times as many cases as SARS did in less than half the time [1]. People with higher education may also be more concerned about national development and social stability, which may make them vulnerable to anxiety during epidemic outbreaks. Many studies have revealed that highly educated individuals tend to suffer from “knowledge anxiety” [45,46]. Furthermore, anxiety can develop into behavioral changes such as constantly disinfecting and scrambling for medicines. Therefore, these results suggest we should not only support medical treatment, but also pay attention to the psychological needs of these subpopulations during pandemics. Media influences mental well-being and can increase anxiety levels [47]. The government should strengthen the core spirit of the media to ensure the validity and accuracy of output information [48]. One report demonstrated that according to data from 2018, there were 112.2 mobile phones per 100 people on average in China, which guaranteed that the government could help people to strengthen their self-protection through online efforts [49]. According to the Compensatory Carry-Over Action Model (CCAM) theory, internet-based interventions and healthy internet activities can be effective at decreasing anxiety and depression among the general population [50]. Therefore, we strongly recommend that the government develop online health education strategies to address mental health issues, promote healthy behavior, and reduce psychological stress. 

Respondents with poor subjective health were vulnerable to anxiety in the multivariate model (β = 2.83, 95% CI: 2.58–3.09). This result was consistent with those reported during the SARS outbreak [51]. The few early deaths reported by the media or by medical experts have indicated that the mortality rate of critically ill COVID-19 patients was related to other common comorbidities and complications. COVID-19 patients with hypertension, cardiac disease, diabetes mellitus, cancer, and COPD were shown to have a higher mortality rate [52,53]. One possible reason is that common comorbidities and complications could reduce body immunity and exacerbate organism damage [54]. People with poor subjective health status did feel they were at high risk of contracting COVID-19, thus increasing their feelings of anxiety.

People with higher risk perception are more likely to experience higher anxiety scores. Since various psychological vulnerability factors will play specific roles in “coronaphobia”, individuals will more easily present diversiform traits such as the intolerance to uncertainty, perceived vulnerability to disease, and susceptibility to anxiety [55]. Once individuals perceive themselves as vulnerable to disease, they will likely experience anxiety, which can leave them unable to tackle the outbreak despite having adequate knowledge, experience, preparation, and resources [11]. Our findings suggest that people with high-risk perception during the epidemic should be targeted for psychological counseling and assistance interventions. Typical clinical mental health consultation requires face-to-face interviews for evaluation. However, face-to-face interviews are challenging in the current scenario where social distancing is necessary to avoid spreading the COVID-19 infection. Therefore, considering online mental health consultation might be more beneficial. Meanwhile, implementing extensive mental health monitoring in the community is worthwhile [56]. 

This study’s results indicated that increased confidence was related to lower levels of anxiety. Vulnerable circumstances caused by the COVID-19 crisis resulted in individual deficiency of psychological resources, thereby lowering people’s self-control and causing some unusual behaviors [57]. One recent study from Japan found that lack of confidence was likely to lead to anxiety [58]. For the general population, adequate confidence during the epidemic is crucial to psychological intervention, which decreases fear and increases mental energy. Based on this finding, positive updates such as scientific information about the epidemic, personal protective measures, and optimistic progress of containment should be made available to the general population in a timely manner.

The anxiety caused by infectious disease potentially yields double-sided results. On the one hand, it may produce damage to individual mental health and further public panic; on the other hand, quick anxiety reactions are an alarm mechanism in humans, a result of millions of years of evolution. It reminds people to pay attention to stress during a dangerous epidemic, when awareness of prevention with early prophylaxis increases. Currently, the inundation of psychological alarm mechanisms is threatening public mental health and well-being. Therefore, future research on anxiety problems triggered by emerging pandemics, especially in the earliest stages, is required. Our findings have certain suggestive significance for Chinese people, and caution should be taken in referencing the conclusion of this study if applied to people of different cultural backgrounds. Future studies should consider cultural aspects of the psychological effects, and repeat our research in other countries with different cultural backgrounds.

## 5. Limitations

There are some limitations to our study. First, online research requires technical competencies related to using the Internet. Second, as causal relationships and attributions are difficult to derive from a cross-sectional analysis, this study was only able to provide preliminary findings. Third, the non-random sampling network survey method potentially caused selection bias, further affecting the results of this study. Due to the nature of an online survey, the sample population was mostly concentrated in urban areas, and often included those with a higher education level. However, we obtained a large sample size, which increased our confidence in our conclusions. Fourth, the study was based on participants’ self-report questionnaires, such that issues of subjectivity and potential bias come into play. Although we conducted quality control, there may be errors in the information. Finally, the procedure of measuring anxiety status can be challenging, which we admit is a limitation of the study. Thus, it can be said that further studies are needed to address these issues and gain more comprehensive knowledge of how to deal with mental health during the pandemic.

## 6. Conclusions 

During the earliest phase of the COVID-19 outbreak, a high proportion of Chinese people were suffering from anxiety. Young females and people with higher education were vulnerable to experiencing anxiety problems during the early days of the outbreak due to the high accessibility of infectious disease information and their ability to judge potential threat. A higher awareness of cognitive risk and poor subjective health contributed to more precariousness, which consequently caused anxiety. On the contrary, individuals who had more confidence in resisting the epidemic had less precariousness, thus leading to lower anxiety risk. Therefore, targeted interventions related to improving public confidence in the containment of the epidemic are necessary for avoiding greater panic. People of Hubei province in China, the epicenter of this disease, were the most anxious. It is necessary to identify people who are vulnerable to decreasing mental health and develop effective intervention strategies to prevent anxiety among them. Evidence from this survey will help provide guidance to policymakers as they create intervention practices during novel disease outbreaks. To address mental health issues during the pandemic, policymakers should adopt several psycho-social interventions to reduce anxiety among the population in affected areas along with methods for controlling outbreaks.

## Figures and Tables

**Figure 1 healthcare-09-00970-f001:**
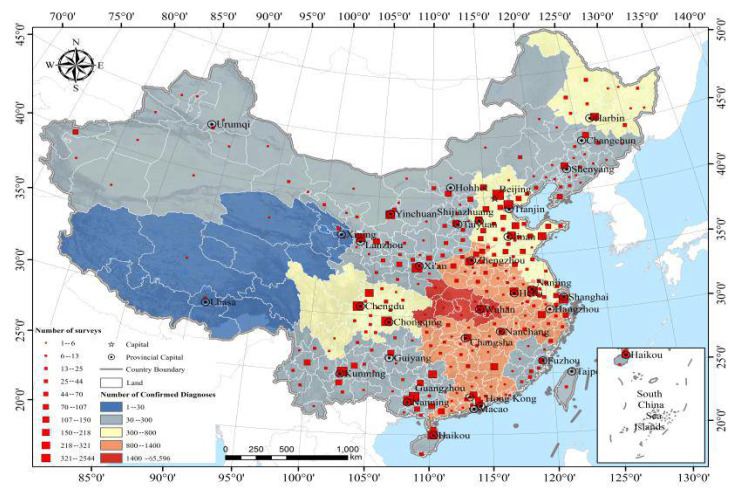
Sample size and number of cases distributed.

**Figure 2 healthcare-09-00970-f002:**
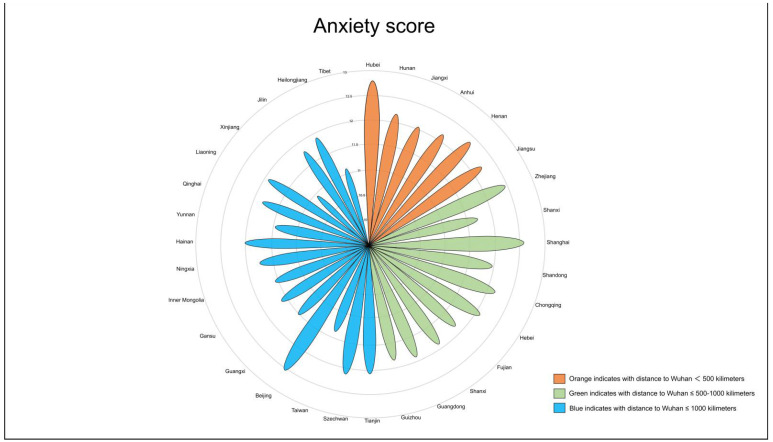
Anxiety scores by province.

**Figure 3 healthcare-09-00970-f003:**
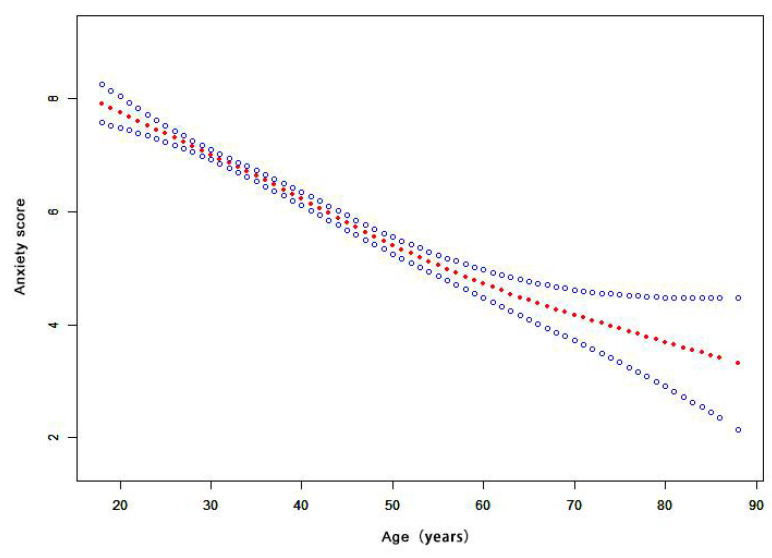
The dose-response relationship of age and anxiety score. (Notes: Generalized additive models; Outcome variable: anxiety score; Exposure variable: age; Adjustment variable: sex, marriage, education, occupation, family members, contact history).

**Table 1 healthcare-09-00970-t001:** Baseline characteristics of people by anxiety score category.

	Normal	Mild Anxiety	Moderate Anxiety	Severe Anxiety	*p* Value
	(*n* = 5490)	(*n* = 2391)	(*n* = 2608)	(*n* = 457)	
**Anxiety score**	2.95 ± 2.09	7.98 ± 0.81	11.04 ± 1.08	14.47 ± 0.50	<0.001
**Age(years)**	38.79 ± 12.61	36.23 ± 12.25	34.57 ± 10.64	32.85 ± 9.90	<0.001
**Sex**					<0.001
Male	2146 (39.09%)	735 (30.74%)	701 (26.88%)	98 (21.44%)	
Female	3344 (60.91%)	1656 (69.26%)	1907 (73.12%)	359 (78.56%)	
**Marriage**					<0.001
Unmarried	1601 (29.16%)	813 (34.00%)	917 (35.16%)	187 (40.92%)	
Married	3664 (66.74%)	1510 (63.15%)	1610 (61.73%)	257 (56.24%)	
Divorced	161 (2.93%)	45 (1.88%)	62 (2.38%)	11 (2.41%)	
Widowed	41 (0.75%)	14 (0.59%)	7 (0.27%)	1 (0.22%)	
Other	23 (0.42%)	9 (0.38%)	12 (0.46%)	1 (0.22%)	
**Education**					<0.001
Senior high school and below	1165 (21.22%)	409 (17.11%)	305 (11.69%)	52 (11.38%)	
Bachelor’s degree	3035 (55.28%)	1395 (58.34%)	1563 (59.93%)	258 (56.46%)	
Master’s degree or above	1290 (23.50%)	587 (24.55%)	740 (28.37%)	147 (32.17%)	
**Occupation**					<0.001
Medical professional	924 (16.83%)	431 (18.03%)	515 (19.75%)	96 (21.01%)	
Laborers	462 (8.42%)	134 (5.60%)	133 (5.10%)	26 (5.69%)	
Teachers and researchers	1129 (20.56%)	437 (18.28%)	452 (17.33%)	61 (13.35%)	
Government staff	195 (3.55%)	88 (3.68%)	129 (4.95%)	18 (3.94%)	
Commercial and service personnel	1093 (19.91%)	488 (20.41%)	509 (19.52%)	77 (16.85%)	
Students	723 (13.17%)	412 (17.23%)	467 (17.91%)	102 (22.32%)	
Retired staff	305 (5.56%)	126 (5.27%)	52 (1.99%)	11 (2.41%)	
Other	659 (12.00%)	275 (11.50%)	351 (13.46%)	66 (14.44%)	
**Residence**					0.002
Urban	4351 (79.25%)	1893 (79.17%)	2145 (82.25%)	382 (83.59%)	
Rural	1139 (20.75%)	498 (20.83%)	463 (17.75%)	75 (16.41%)	
**Area**					0.009
From Hubei province	69 (1.26%)	42 (1.76%)	59 (2.26%)	6 (1.31%)	
From other provinces	5324 (96.98%)	2299 (96.15%)	2485 (95.28%)	444 (97.16%)	
From abroad	97 (1.77%)	50 (2.09%)	64 (2.45%)	7 (1.53%)	
**Family members**					0.270
< 3 family members	566 (10.31%)	210 (8.78%)	263 (10.08%)	43 (9.41%)	
3–5 family members	3085 (56.19%)	1323 (55.33%)	1462 (56.06%)	263 (57.55%)	
≥ 5 family members	1839 (33.50%)	858 (35.88%)	883 (33.86%)	151 (33.04%)	
**Contact history**					<0.001
No	5170 (94.17%)	2203 (92.14%)	2312 (88.65%)	3902 (85.34%)	
Yes	320 (5.83%)	188 (7.86%)	296 (11.35%)	67 (14.66%)	

Continuous variables are expressed by means (±Standard Distribution), and classification variables are expressed as a percentage.

**Table 2 healthcare-09-00970-t002:** The association of age, education, health, and anxiety score in the whole population.

Variables	*n* (%)	Non-Adjusted Model		Model I ^b^		Model II ^c^	
		(95% CI LL, UL) ^a^	*p* Value	(95% CI LL, UL)	*p* Value	(95% CI LL, UL)	*p* Value
**Age category**							
<30 years	3469 (31.69)	0		0		0	
30–40 years	3132 (28.61)	−0.18 (−0.38, 0.02)	0.0704	−0.46 (−0.72, −0.20)	0.0006	−0.40 (−0.68, −0.12)	0.0053
40–50 years	2492 (22.77)	−1.08 (−1.29, −0.87)	<0.0001	−1.28 (−1.57, −0.98)	<0.0001	−1.29 (−1.60, −0.98)	<0.0001
≥50 years	1853 (16.93)	−2.07 (−2.30, −1.85)	<0.0001	−2.15 (−2.46, −1.84)	<0.0001	−2.12 (−2.47, −1.78)	<0.0001
*p* Value for Trend	<0.001						
**Education category**							
Senior high school and below	1931 (17.64)	0		0		0	
Bachelor’s degree	6251 (57.11)	1.21 (1.00, 1.42)	<0.0001	0.89 (0.67, 1.12)	<0.0001	0.84 (0.62, 1.06)	<0.0001
Master’s degree or above	2764 (25.25)	1.55 (1.31, 1.79)	<0.0001	1.19 (0.92, 1.46)	<0.0001	1.15 (0.88, 1.41)	<0.0001
*p* Value for Trend	<0.001						
**Health category**							
Very healthy	6332 (57.85)	0		0		0	
Healthy	3497 (31.95)	1.57 (1.40, 1.73)	<0.0001	1.76 (1.60, 1.92)	<0.0001	1.76 (1.59, 1.92)	<0.0001
Ordinary or unhealthy	1117 (10.20)	2.18 (1.92, 2.43)	<0.0001	2.78 (2.52, 3.03)	<0.0001	2.83 (2.58, 3.09)	<0.0001
*p* Value for Trend	<0.001						

^a^ 95% Confidence Interval (CI) lower limit (LL), upper limit (UL); ^b^ Adjusted for age, sex, and marriage; ^c^ Adjusted for age, sex, marriage, education, occupation, family members, contact history, cognitive risk, confidence, residence, concern.

**Table 3 healthcare-09-00970-t003:** The association of cognitive risk, confidence, and anxiety score in the whole population.

Variables	*n* (%)	Non-Adjusted Model		Model I ^b^		Model II ^c^	
		(95% CI LL, UL) ^a^	*p* Value	(95% CI LL, UL)	*p* Value	(95% CI LL, UL)	*p* Value
**Cognitive risk **							
No risk	1755 (16.01)	0		0		0	
Low risk	6440 (58.80)	2.52 (2.32, 2.72)	<0.0001	2.43 (2.33, 2.63)	<0.0001	2.21 (2.02, 2.41)	<0.0001
Medium risk	1982 (18.17)	4.95 (4.70, 5.19)	<0.0001	4.79 (4.55, 5.03)	<0.0001	4.36 (4.12, 4.60)	<0.0001
High risk	591 (5.40)	4.99 (4.63, 5.34)	<0.0001	4.87 (4.52, 5.22)	<0.0001	4.52 (4.18, 4.87)	<0.0001
Extremely high risk	178 (1.62)	5.27 (4.68, 5.85)	<0.0001	5.15 (4.57, 5.73)	<0.0001	4.89 (4.33, 5.46)	<0.0001
*p* Value for Trend	<0.001						
**Confidence**							
Unconfident	1235 (11.28)	0		0		0	
Somewhat confident	5322 (48.62)	−0.41 (−0.66, −0.16)	0.0012	−0.56 (−0.80, −0.32)	<0.0001	−0.54 (−0.77, −0.31)	<0.0001
Confident	3535 (32.29)	−1.94 (−2.20, −1.68)	<0.0001	−2.00 (−2.26, −1.75)	<0.0001	−1.72 (−1.95, −1.48)	<0.0001
Very confident	854 (7.80)	−2.94 (−3.29, −2.59)	<0.0001	−2.92 (−3.26, −2.58)	<0.0001	−2.45 (−2.77, −2.13)	<0.0001
*p* Value for Trend	<0.001						

^a^ 95% Confidence Interval (CI) lower limit (LL), upper limit (UL); ^b^ Adjusted for age, sex, and marriage; ^c^ Adjusted for age, sex, marriage, education, occupation, family members, contact history, cognitive risk, confidence, residence, concern.

## Data Availability

The data presented in this study are available on request from the corresponding author.

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
