# Peer review of "Who Is the Most Vulnerable to Anxiety at the Beginning of the COVID-19 Outbreak in China? A Cross-Sectional Nationwide Survey"

_healthcare, 2021, doi:10.3390/healthcare9080970_

Round 1
Reviewer 1 Report
In general, the author's research is out of data. There are currently a large number of studies on the impact of the covid-19 epidemic on the anxiety of different groups of people. Although the author has made a more detailed division of different groups, the basic conclusions obtained are not much different from the current research, which makes me feel that the innovation of this paper is not enough.
Line 51: The authors note that there is currently no specific treatment for this severe disease, but the literature cited is from 2020, which does not reflect the current situation.
Line 67: The author did not put forward the research gap very well, but used a simple sentence "To our knowledge..." to illustrate the current deficiencies of the research, which I think is undesirable. In fact, there have been numerous and more detailed studies of mental health problems affected by COVID-19 in different groups. See below:
Rossi, R., Socci, V., Talevi, D., Mensi, S., Niolu, C., Pacitti, F., ... & Di Lorenzo, G. (2020). COVID-19 pandemic and lockdown measures impact on mental health among the general population in Italy. Frontiers in psychiatry, 11, 790.
Dryhurst, S., Schneider, C. R., Kerr, J., Freeman, A. L., Recchia, G., Van Der Bles, A. M., ... & van der Linden, S. (2020). Risk perceptions of COVID-19 around the world. Journal of Risk Research, 23(7-8), 994-1006.
Line 76: Where is Section 2.2? If not, why label section 2.1?
Line 91: When analyzing anxiety, DASS-21 and SASRQ questionnaires are more commonly used. If the author designs a questionnaire here, I think it is necessary to put the questionnaire in the appendix to facilitate other scholars to conduct similar research.
Line 154: In Figure 2, I see that Yunnan, Zhejiang, and Taiwan have the highest anxiety scores.
Related to Figure 2: We know that during the SARS outbreak, Guangzhou, Beijing, Shanxi, Hebei, Tianjin and Jilin were more serious. The previous epidemic will leave people with a psychological shadow. In the conclusion of this article, Shanxi, Hebei, and Tianjin did not show high anxiety. I think it would be very meaningful if the author can explain this phenomenon in the discussion section.
Line 305: This article is still from 2020
Author Response
Response to Reviewer 1 Comments
In general, the author's research is out of data. There are currently a large number of studies on the impact of the covid-19 epidemic on the anxiety of different groups of people. Although the author has made a more detailed division of different groups, the basic conclusions obtained are not much different from the current research, which makes me feel that the innovation of this paper is not enough.
We would like to express our heartfelt gratefulness for your comments and suggestions to the manuscript. We agree with the reviewer.
We have been working long time to analyze the data, decreased the influence of confounding factors, and analyzed the anxiety in women and highly educated people. We believed the findings can still provide reference for later comparison and other studies.
Line 51: The authors note that there is currently no specific treatment for this severe disease, but the literature cited is from 2020, which does not reflect the current situation.
Reply: Thank you very much for pointing this out. We have deleted the sentence and made updates to the second paragraph in introduction section, which was marked in red.
Line 67: The author did not put forward the research gap very well, but used a simple sentence "To our knowledge..." to illustrate the current deficiencies of the research, which I think is undesirable. In fact, there have been numerous and more detailed studies of mental health problems affected by COVID-19 in different groups. See below:
Rossi, R., Socci, V., Talevi, D., Mensi, S., Niolu, C., Pacitti, F., ... & Di Lorenzo, G. (2020). COVID-19 pandemic and lockdown measures impact on mental health among the general population in Italy. Frontiers in psychiatry, 11, 790.
Dryhurst, S., Schneider, C. R., Kerr, J., Freeman, A. L., Recchia, G., Van Der Bles, A. M., ... & van der Linden, S. (2020). Risk perceptions of COVID-19 around the world. Journal of Risk Research, 23(7-8), 994-1006.
Reply: We agreed with this comment. We have made changes, as “Evidence is accumulating to suggest that investigating the level of anxiety in individuals and identifying factors of anxiety can help scholars and practitioners clearly comprehend the severity of the pandemic’s effect and improve the effectiveness of health risk communications[28, 29]. The aim of this study is to evaluate the anxiety reaction of Chinese people and the related determinants during the earliest phase of COVID-19 outbreak in China. Targeted interventions and psychological consultation services can be provided based on the scientific evidence for the target population.” (Page 2, line 85-88, marked in red)
Line 76: Where is Section 2.2? If not, why label section 2.1?
Reply: Thank you for pointing this out. The label was wrong and we have corrected this editing fault as follows, “ 2. Materials and methods ; 2.1 Study population; 2.2 Measures.......”.
Line 91: When analyzing anxiety, DASS-21 and SASRQ questionnaires are more commonly used. If the author designs a questionnaire here, I think it is necessary to put the questionnaire in the appendix to facilitate other scholars to conduct similar research.
Reply: We appreciated the comment. Anxiety reaction was measured via 5-item short forms of the State scale of the Spielberger State-Trait Anxiety Inventory (STAIS-5) and adjusted to adapt Chinese Context. We have attached study questionnaire in the appendix.
Reference: Zsido AN, Teleki SA, Csokasi K, Rozsa S, Bandi SA: Development of the short version of the spielberger state-trait anxiety inventory. Psychiatry research 2020, 291.
Line 154: In Figure 2, I see that Yunnan, Zhejiang, and Taiwan have the highest anxiety scores.
Reply: Terribly sorry for some mistakes made in the Fig2 before. We have modified Figure2 using new standardized.
Related to Figure 2: We know that during the SARS outbreak, Guangzhou, Beijing, Shanxi, Hebei, Tianjin and Jilin were more serious. The previous epidemic will leave people with a psychological shadow. In the conclusion of this article, Shanxi, Hebei, and Tianjin did not show high anxiety. I think it would be very meaningful if the author can explain this phenomenon in the discussion section.
Reply: We appreciated these excellent comments. We have added corresponding illustration in the Discussion section, as “People of Hubei province in China, the epicenter of this disease, were the most anxious, followed by those living in Shanghai, Beijing, Zhejiang, which are some of the most economically prosperous areas in China, leading to the most interaction with Hubei, in turn received the large number of infected travelers returning from Wuhan at the special times that was Chinese New Year, which also was a reason why the most high-risk areas facing outbreak no longer were Guangzhou, Shanxi, Hebei, Tianjin and Jilin though those residents were the most serious during the SARS outbreak. In the early stage of the epidemic, the travel data identified cities and regions susceptible to potential future outbreaks. People living in cities with large population mobility are more prone to anxiety, restlessness, depression and fear. ” (Page 9, line 263-272, marked in red )
Line 305: This article is still from 2020
Reply: Thank you for pointing this out. We have deleted the sentence “The lack of a targeted vaccine.........”
Reviewer 2 Report
The article deals with a very important issue, both from the scientific and practical perspective. The problem of anxiety experienced in connection with the pandemic and its determinants is important for supporting the population in various areas of life, organizing specialist support for risk groups, and planning activities limiting the spread of the pandemic. The material collected by researchers in the course of a well-planned project will be potentially useful from the perspective of future epidemics. My assessment of the project, its concept, and implementation is very positive. Still, I have a few minor comments on the issues that should be supplemented or clarified:
- It is worth expanding the introduction by showing the current state of research on the issue in other countries. This will also be useful in the discussion. I think it is worth mentioning research on experiences accompanying the pandemic carried out in Western countries, including Europe and America. This will create an opportunity to relate in more detail to the cultural dimension of such experiences. Therefore, a question arises: What is the possibility of generalizing the trend to include populations operating in different cultural conditions (e.g. gender-related trends)?
- In the introduction, the authors write, “(...) insufficient numbers of studies focus on adressing and identificating the population groups...”. Please add a relevant reference. The authors also use the phrase "vulnerable groups and specific groups" in this paragraph, referring to their own research. What groups are the authors talking about?
- What are the indices of the "contact story" variable?
- In the research limitations, I suggest adding that online research requires technical competencies related to using the Internet.
Author Response
Response to Reviewer 2 Comments
The article deals with a very important issue, both from the scientific and practical perspective. The problem of anxiety experienced in connection with the pandemic and its determinants is important for supporting the population in various areas of life, organizing specialist support for risk groups, and planning activities limiting the spread of the pandemic. The material collected by researchers in the course of a well-planned project will be potentially useful from the perspective of future epidemics. My assessment of the project, its concept, and implementation is very positive. Still, I have a few minor comments on the issues that should be supplemented or clarified:
Reply: Thank you very much for your positive comments and constructive suggestions, which are really helpful for the improvement of this manuscript.
- It is worth expanding the introduction by showing the current state of research on the issue in other countries. This will also be useful in the discussion. I think it is worth mentioning research on experiences accompanying the pandemic carried out in Western countries, including Europe and America. This will create an opportunity to relate in more detail to the cultural dimension of such experiences. Therefore, a question arises: What is the possibility of generalizing the trend to include populations operating in different cultural conditions (e.g. gender-related trends)?
-
Reply: This is very helpful. We have added the corresponding illustration in the introduction section, as follows: “The lessons learned from the severe acute respiratory syndrome (SARS)[8-10], pandemic influenza A(H1N1)[11-13] and influenza A(H7N9)[14, 15] experiences in various culture settings demonstrate that attention must also be paid to mental health as part of the COVID-19 epidemic prevention and control. Cultural differences, disease perceptions, government involvement and the stage of the outbreak are associated with public response, and these factors vary by diseases and settings[16-18]. ” (Page 2, line 50-60, marked in red ).
As the reviewer pointed, it should be said that this has certain suggestive significance for Chinese people, and caution should be taken in reference to the conclusion of this study if it is from different cultural backgrounds. And we have added the corresponding illustration in the discussion section, as follows: “ Our findings have certain suggestive significance for Chinese people, and caution should be taken in reference to the conclusion of this study if it is from different cultural backgrounds. Future studies should consider culture aspects of the psychological effects, and repeat our research in other countries with different cultural backgrounds.” (Page 11, line 362-367, marked in red ).
- In the introduction, the authors write, “(...) insufficient numbers of studies focus on adressing and identificating the population groups...”. Please add a relevant reference. The authors also use the phrase "vulnerable groups and specific groups" in this paragraph, referring to their own research. What groups are the authors talking about?
-
Reply: Thank you for your suggestion. We have modified the sentence “To our knowledge......” and made minor adjusting to the last paragraph in the Introduction section, as “Evidence is accumulating to suggest that investigating the level of anxiety in individuals and identifying factors of anxiety can help scholars and practitioners clearly comprehend the severity of the pandemic’s effect and improve the effectiveness of health risk communications[28, 29]. The aim of this study is to evaluate the anxiety reaction of Chinese people and the related determinants during the earliest phase of COVID-19 outbreak in China. Targeted interventions and psychological consultation services can be provided based on the scientific evidence for the target population.” (Page 2, line 85-92, marked in red )
- What are the indices of the "contact story" variable?
-
Reply: Thank you for pointing this out. We have added a detailed description of the “contact history” variable in the Methods section, as follows, “ Contact history variables included close contact with an individual with confirmed COVID-19, indirect contact with an individual with confirmed COVID-19, and contact with an individual with suspected COVID-19 or infected materials.” (Page 3, line 109-112, marked in red )
- In the research limitations, I suggest adding that online research requires technical competencies related to using the Internet.
-
Reply: We appreciated this comment. We have added this in Limitation part , as follows. “ First, online research requires technical competencies related to using the Internet......”(Limitations part, Line 369-370, marked in red)

Reviewer 3 Report
Aims and scope: the manuscript falls within the subject area of the Healthcare journal.
Title: the title complies with the requirements of the journal.
Abstract: the abstract partially complies with the editorial recommendations – is qualitatively correct but quantitatively to long. „(…) to identify associations between some factors and anxiety scores.” – what factors? (lines 16-17).
Inrtoduction: „By the end of June 2020, more than 10 million confirmed COVID-19 cases were detected in 216 countries, territories, and areas, and more than 500,000 deaths had been reported” – it needs references (lines 38-39). „A few studies have begun focusing on all aspects of the disease since the outbreak, but there is currently no specific treatment for this severe condition [9]” – the word few means more than one, so the authors should provide more than one reference. The authors should describe in more detail why they focused on early phase of the COVID-19 pandemic (lines 47-50). What preparation did the authors have in mind? (lines 49-50).
Materials and Methods: how was nationality verified in relation to foreign respondents? (line 79). Since almost 11,000 online questionnaires were collected during one week, why did the authors not extend the duration of the study, e.g. by another week in order to increase the representativeness of the sample? Were psychometric indicators calculated for non-standardized questions? How was the age of the respondents verified? – respondents under the age of 18 were rejected (line 130). What hypotheses were verified?
Results: the results are properly presented, legible and understandable.
Discussion: this section is properly compiled.
Conclusions: The conclusions resemble the results too much – they should constitute a kind of extension of the interpretation of the obtained results. It would be necessary to go a step further and answer the question: what does the result show?
References: a significant numbers of articles was published after 2015. This section contains minor glitches according to requirements of the journal.
General comment: the manuscript has been carefully compiled and reading it is not a problem. Good job! There are some shortcomings in the manuscript – punctuation and stylistic errors.
Author Response
Response to Reviewer 3 Comments
Aims and scope: the manuscript falls within the subject area of the Healthcare journal.
Reply: Thank you for your supporting.
Title: the title complies with the requirements of the journal.
Abstract: the abstract partially complies with the editorial recommendations – is qualitatively correct but quantitatively to long. „(…) to identify associations between some factors and anxiety scores.” – what factors? (lines 16-17).
Reply: Thank you for pointing this out. We have added detailed description.(Page 1, line 20, marked in red )
Introduction: „By the end of June 2020, more than 10 million confirmed COVID-19 cases were detected in 216 countries, territories, and areas, and more than 500,000 deaths had been reported” – it needs references (lines 38-39). „A few studies have begun focusing on all aspects of the disease since the outbreak, but there is currently no specific treatment for this severe condition [9]” – the word few means more than one, so the authors should provide more than one reference. The authors should describe in more detail why they focused on early phase of the COVID-19 pandemic (lines 47-50). What preparation did the authors have in mind? (lines 49-50).
Reply: We appreciated these suggestions. The details of our modifications are provided below:
1). We have added the references to “By the end of June 2020, more than 10 million confirmed COVID-19 cases were detected in 216 countries, territories, and areas, and more than 500,000 deaths had been reported”
2) Due to the “there is currently no specific......” does not reflect the current situation, we have updated the sentence and made minor adjustments to the second paragraph in introduction section.
3). We have added detailed illustration to explain why they focused on early phase of the COVID-19 pandemic in the introduction section, as “In the early phase of pandemic, the lack of knowledge on the virus and the absence of valid information is easy to cause universal anxiety and panic among general population[19]. Examining the general population’s psychological response during the initial phase of an emerging epidemic is useful to inform both policy makers and the public about the state of preparedness.”(Page 2, line 56-60, marked in red )
Materials and Methods: how was nationality verified in relation to foreign respondents? (line 79). Since almost 11,000 online questionnaires were collected during one week, why did the authors not extend the duration of the study, e.g. by another week in order to increase the representativeness of the sample? Were psychometric indicators calculated for non-standardized questions? How was the age of the respondents verified? – respondents under the age of 18 were rejected (line 130). What hypotheses were verified?
Reply: As the aim of the study was to measure people’s anxiety state at one week after the Wuhan lockdown, the online survey time range was set to one week. We did calculate the sample size, and the study sample size (N=11000) was adequate. People under 18 years old were not within the scope of our study because parental consent would have been required. In addition, the online survey may suffer the information bias, we could not exclude the foreign respondents, but assumed the sample of foreign respondents is small and it would not change our conclusion since we used the Chinese questionnaire and included the respondents only in China by validating the IP address. Furthermore, we did calculate the psychometric indicators for non-standardized questions.
Results: the results are properly presented, legible and understandable.
Reply: Thank you for your supporting.
Discussion: this section is properly compiled.
Reply: Thank you for your supporting.
Conclusions: The conclusions resemble the results too much – they should constitute a kind of extension of the interpretation of the obtained results. It would be necessary to go a step further and answer the question: what does the result show?
Reply: We appreciated this comment. We have added further illustration in the Conclusion section, as follows “ During the earliest phase of the COVID-19 outbreak, a high proportion of Chinese people were suffering from anxiety. Young females and people with higher education were vulnerable to experiencing anxiety problems during the early days of the outbreak due to high accessibility of infectious disease information and judge ability of threat. People with higher awareness of cognitive risk and poor subjective health contribute to more precariousness, which was prone to cause anxiety. On the contrary, individuals who had more confidence in resisting the epidemic had less precariousness, thus leading to lower anxiety risk. So, targeted interventions related to improving public confidence on containing the epidemic are necessary for avoiding greater panic. People of Hubei province in China, the epicenter of this disease, were the most anxious. It is necessary to identify people who are vulnerable to decreasing mental health and develop effective intervention strategies to prevent anxiety among them. Evidence from this survey will help provide guidance to policymakers as they create intervention practices during novel disease outbreaks. To address mental health issues during the pandemic, policymakers should adopt several psycho-social interventions to reduce anxiety among the population in affected areas along with methods of controlling outbreaks.” (Page 12, line 385-392, marked in red )
References: a significant numbers of articles was published after 2015. This section contains minor glitches according to requirements of the journal.
Reply: Many thanks for the suggestion, we have updated references in the text. (see References list)
General comment: the manuscript has been carefully compiled and reading it is not a problem. Good job! There are some shortcomings in the manuscript – punctuation and stylistic errors.
Reply: Thank you very much for your positive comments. We have corrected those punctuation and stylistic errors.

Reviewer 4 Report
This is a valuable study on anxiety levels in different groups residing in China. While the findings are really important, the study has some limitations which need to be addressed:
1) explain in the abstract, WHY this research is important, and why it is key to understand to evaluate the anxiety reaction of Chinese people and the related determinants during the earliest phase of 11 COVID-19 outbreak in China.
2) add a theoretical backdrop to your study and describe it in the intro as well as relate your findings to it in the discussion. Such a theory can be the CCAM, see doi: 10.2147/PRBM.S274502
3) while I agree that too much anxiety is bad, some low levels are actually required to initiate awareness, thus I strongly recommend to clearly define "healthy levels of anxiety" (was this the low group?), "elevated but not problematic level of a." (was this the medium group?), and "problematic level of a." (was this the high group?) and rerun the analysis with differentiating this in a more practically/clinical relevant framework.
4) explain what you actually mean with "population groups that are vulnerable to mental health problems"
5) Why did you not use for "Subjective health status" a validated measurement like the single item from the SF12?
6) Cognitive risk needs to be introduced and explained in the introduction section sufficiently: Why do you think this is important and distinct from anxiety?
7) Where study participants compensated for their participation in the study?
8) how did you deal with missing values?
9) run a power analysis (also see my following point in this regard)
10) With your results in Tab 1, I fear you found everything statistically significant only due to your large sample size. Especially with Area and Family Member I think this is an error and alpha error adjustment is required.
11) Fig 2 is nice but the reader requires this info related to infection rates in these regions or with distance to Wuhan. Maybe you can reorder the regions in an ordered way and add this additional information?
12) Tab 2 and 3, there are relatively small cell sizes which lowers the reliability of the results i.e. the results appears not trustworthy. Thus, I suggest combining the
-age groups <20 with 20-30y; 50-60y with 60+
- junior high school and below with senior high school
- health category ordinary with unhealthy+
- confidence: very unconf. with unconf.
13)under author contrib. you speak of an experiment but in the paper you only report the survey data. pls explain or correct
14) add notes to the figures in the appendix to explain what you actually display. consider moving the figure with age into the main body text.
Author Response
Response to Reviewer 4 Comments
This is a valuable study on anxiety levels in different groups residing in China. While the findings are really important, the study has some limitations which need to be addressed:
Thank you very much for your positive comments and constructive suggestions, which are really helpful for the improvement of this manuscript. We have dealt with these limitations and illustrated our replies as follows:
1) explain in the abstract, WHY this research is important, and why it is key to understand to evaluate the anxiety reaction of Chinese people and the related determinants during the earliest phase of 11 COVID-19 outbreak in China.
Reply: We appreciated this comments, we have added the corresponding illustration as follows: “The COVID-19 pandemic has not only changed the people’s health behavior, but also induced the psychological reaction among the public. Research data is needed to develop scientific evidence-driven strategies to reduce adverse mental health effects. The aims of this study are to evaluate the anxiety reaction of Chinese people and the related determinants during the earliest phase of COVID-19 outbreak in China. Evidences from this survey will contribute to a targeted reference to how to deliver psychological counseling service facing outbreaks.” (Page 1, line 10-11, line 13-15, marked in red )
2) add a theoretical backdrop to your study and describe it in the intro as well as relate your findings to it in the discussion. Such a theory can be the CCAM, see doi: 10.2147/PRBM.S274502
Reply: This is a helpful comment. We have studied recommended carefully, and added corresponding illustration in the discussion section, as: “According to the Compensatory Carry-Over Action Model (CCAM) theory, internet-based interventions and healthy internet activities can be effective at decreasing anxiety and depression among general population [51].” (Page10, line 315-317, marked in red )
3)while I agree that too much anxiety is bad, some low levels are actually required to initiate awareness, thus I strongly recommend to clearly define "healthy levels of anxiety" (was this the low group?), "elevated but not problematic level of a." (was this the medium group?), and "problematic level of a." (was this the high group?) and rerun the analysis with differentiating this in a more practically/clinical relevant framework.
Reply: We appreciated these excellent comments. According to the 5-item short forms of the State scale of the Spielberger State-Trait Anxiety Inventory(STAIS-5), the total anxiety score was divided into normal(0-6), mild anxiety(7-9), moderate anxiety(10-13), severe anxiety(14-15). Someone scoring ≥10 on the STAIS-5 should be considered potentially clinically anxious.
Reference: Zsido AN, Teleki SA, Csokasi K, Rozsa S, Bandi SA: Development of the short version of thespielberger state-trait anxiety inventory. Psychiatry research 2020, 291.
4) explain what you actually mean with "population groups that are vulnerable to mental health problems"
Reply: This is helpful. We have deleted the sentence to avoid the confusion induced.
5) Why did you not use for "Subjective health status" a validated measurement like the single item from the SF12?
Reply: Thank you for your suggestion. Considering ease of operation and easy of data collection, we did not use a validated measurement like the single item from the SF12 but rather quantitative measure. This would be a small limitation in study design. We will try to eliminate these drawbacks in our next study using validated measurement tool.
6) Cognitive risk needs to be introduced and explained in the introduction section sufficiently: Why do you think this is important and distinct from anxiety?
Reply: This is a helpful comment. We have added the corresponding illustration in the Introduction section, as follows: “Thus, an pessimism of any kind of information can have negatively psychological effects, easy leading public cognitive risk, which is an inherent feature to all human cognitive activity, as well as an index that can measure the psychological panic of the public. Previous studies suggested that people with higher risk perceptions were more likely to take comprehensive precautionary measures against infection[21, 22]. At the same time, risk cognition also affects public psychology states. Excessive risk cognition to be increased the likelihood of an array of negative emotions including anxiety and panic[23].” (Page 2, line 66-73, marked in red )
7) Where study participants compensated for their participation in the study?
Reply: This study was approved by ethnic review committee. Participation was anonymous, voluntary and unpaid.
8) how did you deal with missing values?
Reply: Thank you for pointing this out. Overall, the data quality of the study was relatively good. Missing data were processed using the listwise deletion method, and the entire sample was excluded from analysis if any single value was missing.
9) run a power analysis (also see my following point in this regard)
Reply: We have reanalyzed the data according to your suggestions.
10) With your results in Tab 1, I fear you found everything statistically significant only due to your large sample size. Especially with Area and Family Member I think this is an error and alpha error adjustment is required.
Reply: We agreed with this comment. Large sample size indeed may result in false positive. However, these demographic variables were included in the current study as covariates. These covariates did not affect our main results and conclusions. We have corrected the errors, but “Family Member” still not significant.
11) Fig 2 is nice but the reader requires this info related to infection rates in these regions or with distance to Wuhan. Maybe you can reorder the regions in an ordered way and add this additional information?
Reply: This is very helpful. Terribly sorry for some mistakes made in the Fig2 before. We have modified Figure2 using new standardized. For much more visualized contrast, Fig2 was changed to lobe pattern. And we have added the distance information in Fig2.
12) Tab 2 and 3, there are relatively small cell sizes which lowers the reliability of the results i.e. the results appears not trustworthy. Thus, I suggest combining the
-age groups <20 with 20-30y; 50-60y with 60+
- junior high school and below with senior high school
- health category ordinary with unhealthy+
- confidence: very unconf. with unconf.
Reply: Thank you for your suggestion. We have combined the groups (age groups <20 with 20-30y; 50-60y with 60+; junior high school and below with senior high school; health category ordinary with unhealthy+; very unconf. with unconf). For the combined groups, we have reanalyzed the data, as Table1; Table2; Table3.
13) under author contrib. you speak of an experiment but in the paper you only report the survey data. pls explain or correct
Reply: Thank you for pointing this out. We have corrected these editing faults.(Page 12, line 405, marked in red )
14) add notes to the figures in the appendix to explain what you actually display. consider moving the figure with age into the main body text.
Reply: We have added notes to the appendix figures and moved the age dose-response curve to the main body text, labeled Figure3.
Round 2
Reviewer 1 Report
This version is much better than the previous one, but I reserve my opinion on the lack of innovation in the article.
Author Response
This version is much better than the previous one, but I reserve my opinion on the lack of innovation in the article.
Reply: Thank you very much for your recognition of our revision. In terms of innovation, we need to give some clarification.
We used “anxiety” and “coronavirus disease 2019” as keywords to search the Medline database. 185 articles were identified after title and abstract screening, of which 23 similar articles were included for comparison. Among the 23 included articles, 5 were Chinese population studies[1-5] and 18 were non-Chinese population studies[6-23].
The 5 Chinese population papers have following features in common:
1)The sample sizes are modest (N= 1011; 2651; 1210; 1104; 1638).
2)Lack of area anxiety level data.
3)Although explored some factors impacting on anxiety reaction, these are not comprehensive enough.
Taken together, compared to above similar studies, our study has three strengths:
1) Our research is the large sample size (N=10946) covered all 31 provinces of China,which was found to describe the anxiety level of Chinese population well.
2) We separately reported anxiety levels of each province in China.
3) Five risk factors related to experiencing anxiety were identified in our study, including young, female, higher education, higher awareness of cognitive risk, poor subjective health. Individual confidence in resisting the epidemic was protective factor.
Thus, our study can provide some scientific evidence and decision-making references. Thank you again.
23 included articles:
- Qian, M.; Wu, Q.; Wu, P.; Hou, Z.; Liang, Y.; Cowling, B.J.; Yu, H. Anxiety levels, precautionary behaviours and public perceptions during the early phase of the COVID-19 outbreak in China: a population-based cross-sectional survey. BMJ Open 2020, 10, e040910, doi:10.1136/bmjopen-2020-040910.
- Shi, Z.; Qin, Y.; Chair, S.Y.; Liu, Y.; Tian, Y.; Li, X.; Hu, W.; Wang, Q. Anxiety and depression levels of the general population during the rapid progressing stage in the coronavirus disease 2019 outbreak: a cross-sectional online investigation in China. BMJ Open 2021, 11, e050084, doi:10.1136/bmjopen-2021-050084.
- Wang, C.; Pan, R.; Wan, X.; Tan, Y.; Xu, L.; Ho, C.S.; Ho, R.C. Immediate Psychological Responses and Associated Factors during the Initial Stage of the 2019 Coronavirus Disease (COVID-19) Epidemic among the General Population in China. Int J Environ Res Public Health 2020, 17, doi:10.3390/ijerph17051729.
- Liang, Y.; Wu, K.; Zhou, Y.; Huang, X.; Zhou, Y.; Liu, Z. Mental Health in Frontline Medical Workers during the 2019 Novel Coronavirus Disease Epidemic in China: A Comparison with the General Population. Int J Environ Res Public Health 2020, 17, doi:10.3390/ijerph17186550.
- Yang, X.; Xiong, Z.; Li, Z.; Li, X.; Xiang, W.; Yuan, Y.; Li, Z. Perceived psychological stress and associated factors in the early stages of the coronavirus disease 2019 (COVID-19) epidemic: Evidence from the general Chinese population. PLoS One 2020, 15, e0243605, doi:10.1371/journal.pone.0243605.
- Islam, M.S.; Ferdous, M.Z.; Potenza, M.N. Panic and generalized anxiety during the COVID-19 pandemic among Bangladeshi people: An online pilot survey early in the outbreak. J Affect Disord 2020, 276, 30-37, doi:10.1016/j.jad.2020.06.049.
- Goularte, J.F.; Serafim, S.D.; Colombo, R.; Hogg, B.; Caldieraro, M.A.; Rosa, A.R. COVID-19 and mental health in Brazil: Psychiatric symptoms in the general population. J Psychiatr Res 2021, 132, 32-37, doi:10.1016/j.jpsychires.2020.09.021.
- Mautong, H.; Gallardo-Rumbea, J.A.; Alvarado-Villa, G.E.; Fernández-Cadena, J.C.; Andrade-Molina, D.; Orellana-Román, C.E.; Cherrez-Ojeda, I. Assessment of depression, anxiety and stress levels in the Ecuadorian general population during social isolation due to the COVID-19 outbreak: a cross-sectional study. BMC Psychiatry 2021, 21, 212, doi:10.1186/s12888-021-03214-1.
- Jungmann, S.M.; Witthöft, M. Health anxiety, cyberchondria, and coping in the current COVID-19 pandemic: Which factors are related to coronavirus anxiety? J Anxiety Disord 2020, 73, 102239, doi:10.1016/j.janxdis.2020.102239.
- Hazarika, M.; Das, S.; Bhandari, S.S.; Sharma, P. The psychological impact of the COVID-19 pandemic and associated risk factors during the initial stage among the general population in India. Open J Psychiatry Allied Sci 2021, 12, 31-35, doi:10.5958/2394-2061.2021.00009.4.
- Hassannia, L.; Taghizadeh, F.; Moosazadeh, M.; Zarghami, M.; Taghizadeh, H.; Dooki, A.F.; Fathi, M.; Alizadeh-Navaei, R.; Hedayatizadeh-Omran, A.; Dehghan, N. Anxiety and Depression in Health Workers and General Population During COVID-19 in IRAN: A Cross-Sectional Study. Neuropsychopharmacol Rep 2021, 41, 40-49, doi:10.1002/npr2.12153.
- Moghanibashi-Mansourieh, A. Assessing the anxiety level of Iranian general population during COVID-19 outbreak. Asian J Psychiatr 2020, 51, 102076, doi:10.1016/j.ajp.2020.102076.
- Fiorillo, A.; Sampogna, G.; Giallonardo, V.; Del Vecchio, V.; Luciano, M.; Albert, U.; Carmassi, C.; Carrà, G.; Cirulli, F.; Dell'Osso, B.; et al. Effects of the lockdown on the mental health of the general population during the COVID-19 pandemic in Italy: Results from the COMET collaborative network. Eur Psychiatry 2020, 63, e87, doi:10.1192/j.eurpsy.2020.89.
- Nagasu, M.; Muto, K.; Yamamoto, I. Impacts of anxiety and socioeconomic factors on mental health in the early phases of the COVID-19 pandemic in the general population in Japan: A web-based survey. PLoS One 2021, 16, e0247705, doi:10.1371/journal.pone.0247705.
- Naser, A.Y.; Dahmash, E.Z.; Al-Rousan, R.; Alwafi, H.; Alrawashdeh, H.M.; Ghoul, I.; Abidine, A.; Bokhary, M.A.; Al-Hadithi, H.T.; Ali, D.; et al. Mental health status of the general population, healthcare professionals, and university students during 2019 coronavirus disease outbreak in Jordan: A cross-sectional study. Brain Behav 2020, 10, e01730, doi:10.1002/brb3.1730.
- Massad, I.; Al-Taher, R.; Massad, F.; Al-Sabbagh, M.Q.; Haddad, M.; Abufaraj, M. The impact of the COVID-19 pandemic on mental health: early quarantine-related anxiety and its correlates among Jordanians. East Mediterr Health J 2020, 26, 1165-1172, doi:10.26719/emhj.20.115.
- Zarrouq, B.; Abbas, N.; Hilaly, J.E.; Asri, A.E.; Abbouyi, S.; Omari, M.; Malki, H.; Bouazza, S.; Moutawakkil, S.G.; Halim, K.; et al. An investigation of the association between religious coping, fatigue, anxiety and depressive symptoms during the COVID-19 pandemic in Morocco: a web-based cross-sectional survey. BMC Psychiatry 2021, 21, 264, doi:10.1186/s12888-021-03271-6.
- Muhammad Alfareed Zafar, S.; Junaid Tahir, M.; Malik, M.; Irfan Malik, M.; Kamal Akhtar, F.; Ghazala, R. Awareness, anxiety, and depression in healthcare professionals, medical students, and general population of Pakistan during COVID-19 Pandemic: A cross sectional online survey. Med J Islam Repub Iran 2020, 34, 131, doi:10.34171/mjiri.34.131.
- Passos, L.; Prazeres, F.; Teixeira, A.; Martins, C. Impact on Mental Health Due to COVID-19 Pandemic: Cross-Sectional Study in Portugal and Brazil. Int J Environ Res Public Health 2020, 17, doi:10.3390/ijerph17186794.
- Alkhamees, A.A.; Alrashed, S.A.; Alzunaydi, A.A.; Almohimeed, A.S.; Aljohani, M.S. The psychological impact of COVID-19 pandemic on the general population of Saudi Arabia. Compr Psychiatry 2020, 102, 152192, doi:10.1016/j.comppsych.2020.152192.
- Alamri, H.S.; Algarni, A.; Shehata, S.F.; Al Bshabshe, A.; Alshehri, N.N.; AM, A.L.; Hussain, A.H.; Alalmay, A.Y.; Alshehri, E.A.; Alqarni, Y.; et al. Prevalence of Depression, Anxiety, and Stress among the General Population in Saudi Arabia during Covid-19 Pandemic. Int J Environ Res Public Health 2020, 17, doi:10.3390/ijerph17249183.
- Rodríguez-Rey, R.; Garrido-Hernansaiz, H.; Collado, S. Psychological Impact and Associated Factors During the Initial Stage of the Coronavirus (COVID-19) Pandemic Among the General Population in Spain. Front Psychol 2020, 11, 1540, doi:10.3389/fpsyg.2020.01540.
- Pisano, S.; Catone, G.; Gritti, A.; Almerico, L.; Pezzella, A.; Santangelo, P.; Bravaccio, C.; Iuliano, R.; Senese, V.P. Emotional symptoms and their related factors in adolescents during the acute phase of Covid-19 outbreak in South Italy. Ital J Pediatr 2021, 47, 86, doi:10.1186/s13052-021-01036-1.
Reviewer 4 Report
The authors did a great job revising their manuscript on basis of my feedback. However, they should carefully proofread it again and make sure that the INVALID CITATIONs are corrected. Otherwise, I can just congratulate your for such valuable work!
Author Response
The authors did a great job revising their manuscript on basis of my feedback. However, they should carefully proofread it again and make sure that the INVALID CITATIONs are corrected. Otherwise, I can just congratulate your for such valuable work!
Reply: Thank you again for your positive and constructive comments on our manuscript. We have checked the manuscript carefully and corrected the references mistakes. (Page 18-19, Reference section, Ref 49; Ref 53; Ref 54, marked in red)